# Investigation of the Suitability of a Commercial Radiation Sensor for Pretherapy Dosimetry of Radioiodine Treatment Patients

**DOI:** 10.3390/s22239392

**Published:** 2022-12-01

**Authors:** Janet O’Callaghan, Dervil Cody, Jennie Cooke

**Affiliations:** 1Medical Physics and Bioengineering Department, St. James’s Hospital Dublin, D08 NHY1 Dublin, Ireland; 2Centre for Industrial and Engineering Optics, School of Physics, Clinical and Optometric Sciences, College of Sciences and Health, Technological University Dublin, D07 EWV4 Dublin, Ireland; 3Medical Physics Department, Children’s Health Ireland at Crumlin, D12 N512 Dublin, Ireland

**Keywords:** radiation sensor, dosimetry, smartphone operated sensor, radioiodine therapy

## Abstract

Radioiodine (I-131) therapy is routinely used to treat conditions of the thyroid. Dosimetry planning in advance of I-131 therapy has been shown to improve patient treatment outcomes. However, this pretherapy dosimetry step requires multiple outpatient appointments and is not feasible for patients living at greater distances. Here, the feasibility of a commercially available smartphone-operated radiation sensor (Smart Geiger Pro, Technonia) for at-home patient pretherapy dosimetry has been investigated. The influence of both treatment-specific parameters (radioisotope activity, gamma photon energy, patient size) and external factors (sensor placement and motion) on the ability of the radiation sensor to accurately quantify radiation dose rates has been studied. The performance limits of the radiation sensor have been identified. A preliminary trial of the sensor on four I-131 patients prior to their therapy, conducted at the Nuclear Medicine/Endocrinology departments of St James’s Hospital Dublin, is also presented. A comparable performance between the low-cost radiation sensor and that of a hospital-grade thyroid uptake probe is reported. This work demonstrates the potential of low-cost commercially available radiation sensors as a solution for at-home pretherapy dosimetry for long distance patients, or indeed for hospitals who wish to implement dosimetry at reduced cost. Recommended conditions for optimum sensor performance use are presented.

## 1. Introduction

Hyperthyroidism is the consequence of excessive thyroid hormone action. Symptoms include nervousness, depression, palpitations, tremors, and impaired fertility [1,2,3]. Therapeutic options for patients with hyperthyroidism include treatment with anti-thyroid drugs, surgical procedures such as thyroidectomy, and radioiodine therapy. Iodine (I)-131 is a radioisotope with a half-life of 8.02 days which decays by beta (606 keV; 90%) and gamma photon (364 keV; 10%) emission. I-131 is selected for treatment of hyperthyroidism and certain types of thyroid cancer because of its effectiveness at inducing cell death; it is also highly penetrative and can penetrate cells up to several millimeters away [4]. Unlike surgical thyroidectomy procedures, radioiodine therapy of the thyroid is non-invasive. While it necessitates the use of ionizing radiation, a 2020 study of 16,637 patients who had undertaken radioiodine treatment for hyperthyroidism, found that there was no association between administration of I-131 treatment for hyperthyroidism and increased risk of incident cancers during a long-term follow-up [5]. Standard radioiodine therapy strategies involve administration of a fixed activity of I-131 to all patients; this approach has the benefits of cost-effectiveness and simplicity. However, according to the British Thyroid Foundation, more than 60% of patients will develop hypothyroidism (i.e., insufficient thyroid action) following standard radioiodine therapy [6].

Radioiodine treatment of hyperthyroidism and benign thyroid disease is the quintessence of nuclear medicine therapy, but the question of the ‘optimal’ radioiodine dose to administer still lacks a definitive answer [7]. The objective of dosimetry prior to therapy is to determine the patient-specific I-131 activity that is most likely to lead to therapeutic success, while limiting the radiation exposure to the amount necessary [8]. This is in keeping with the ALARA principle in radiation protection, which states that a patient’s radiation dose should be kept ‘as low as reasonably achievable’. It is also mandated by the Basic Safety Standards Directive of the European Union [9]. Despite this, there has been no widespread adoption of this approach among centers offering I-131 therapy for benign thyroid diseases. Pre-therapy dosimetry in I-131 therapy requires the measurement of the patient-specific uptake of a low-dose of radioiodine prior to therapy, and analysis of changes in radioactivity over time [10]. Individual patient dosimetry is essential both for optimizing the administered activity through the establishment of minimum effective and maximum tolerated absorbed doses, and for determining a dose–response relationship as a basis for predicting thyroid gland response. A 2018 retrospective study of radioiodine therapy patients reported that the use of a personalized dosimetric therapy delayed the long-term onset of hypothyroidism in 26% of patients; this was achieved using much lower administrated activities than is currently recommended [11]. While robust, randomized clinical data on the effectiveness of personalized dosimetry in human radiation therapy patients is scarce [10], a 2021 study of 1,688 hyperthyroid cats demonstrated that the use of an individualized dose based on pretherapy measurements resulted in cure rates similar to historical treatment rates, despite the administration of much lower I-131 treatment doses [12]. In the feline-based study, identification of the ‘sufficient’ dose of I-131 to administer required extensive pretherapy measurements of serum thyroid hormone concentrations, quantitative thyroid scintigraphy, and the percentage I-131 uptake by the thyroid.

Pretherapy dosimetry for outpatient I-131 therapy is in use in the Nuclear Medicine/Endocrinology departments of St. James’s Hospital Dublin. The pretherapy involves the administration of an I-131 diagnostic capsule with a known activity of typically between 0.5–5 MBq. The radioactivity retained in the thyroid gland is then measured intermittently at approximately 24, 48 and 96 h after administration via the use of a hospital-grade thyroid uptake radiation sensor. The patient specific radioiodine uptake (*RIU*) as a function of time, *t*, can be calculated using Equation (1):(1)$$RIU\;\left(t\right)=\frac{Net\;counts\;from\;patient\;thyroid\;gland}{Net\;counts\;administered}$$

In order to obtain the *net counts administered* reference value, a second I-131 capsule with identical activity is prepared at the same time as the patient capsule. This one is dissolved and placed in a thyroid uptake neck phantom (Figure 1a) to simulate typical attenuation of the neck but with 100% uptake. The uptake curve as a function of time allows for the calculation of the optimum activity for an individualized therapy. This personalized dosimetry service was offered to all patients attending St. James’s Hospital for radioiodine therapy as treatment for hyperthyroidism. However, it was found that approximately 1 in 3 patients were unable to undergo dosimetry due to the multiple outpatient appointments and travel to the hospital required. Even though pretherapy treatment planning is of benefit to patients, the disincentivizing burden of multiple appointments and travel on radiation therapy patients is widely acknowledged [10,13,14].

In recent years, efforts have been made to develop wearable radiation detectors to enable remote monitoring of I-131 uptake in the thyroid following administration of the I-131 therapy; however, no studies to date have focused on pretherapy dosimetry itself. In 2017, Brinks et al. reported their development of a neck collar-based detector device as a potential method for remote I-131 uptake monitoring which could ease the burden on radioiodine patients [15]. In this study, the novel measurement device was used during therapy (initial administered I-131 activity of 500 MBq) to measure the patient’s I-131 uptake curve, and the counts per second were displayed on an associated handheld tablet. While the proposed collar-based device was deemed suitable for such measurements, patients reported discomfort while wearing the collar for extended periods. This suggests that a patient self-monitoring device should ideally be small, handheld, and lightweight. The feasibility of a similar collar-based radiation detector for use at home by patients who have undergone radioiodine therapy has also recently been reported [16]. As in [15], the intended purpose of this device is to monitor the real-time uptake of I-131 by the thyroid, once the I-131 treatment has been administered, rather than for pretherapy dosimetry applications.

Here, the suitability and feasibility of a commercially available, low-cost smartphone-operated gamma radiation sensor (Smart Geiger Pro, Technonia [17]) for remote pre-I-131 therapy dosimetry is reported. Due to its lightweight nature and simplistic mode of operation, the device has the potential to be operated by a patient in their own home without supervision, removing the need for multiple outpatient visits. Furthermore, the handheld nature of the device, rather than a wearable collar, ensures minimal patient discomfort during the measurement, which is crucial to maximize patient compliance. The low cost and commercial availability of the sensor is essential to enable its potential use by medical professionals in a variety of settings. The aim of this study was to explore the functionality and performance of this radiation sensor, to determine if it is reliable and sensitive enough to be used by patients for at home I-131 pretherapy uptake monitoring. To the best of the authors’ knowledge, this is the first time a feasibility study of a radiation sensor for pre-I-131 therapy dosimetry has been reported. The influence of a range of parameters on the Smart Geiger Pro (SGP) sensor performance were investigated. These parameters can be loosely divided into two categories: (1) treatment parameters, which are governed by the radioisotope used and patient size (i.e., radioisotope activity, photon energy, influence of scattering material/tissue); and (2) external parameters, which are practically determined by the patient’s method of using the sensor (motion and positioning of the sensor during use, acquisition/count time used). A feasibility study consisting of four patients was conducted in St. James’s Hospital Dublin; this involved comparative measurements of RIU obtained using both the SGP sensor and a standard thyroid uptake sensor. Recommendations for use of the SGP sensor are discussed.

## 2. Materials and Methods

### 2.1. Description and Operation of the Smart Geiger Pro Sensor

The SGP (model number: SGP-001) radiation sensor was purchased from Technonia. It is lightweight (13.5 g) and relatively small with dimensions of 75 mm × 24 mm. Limited information is available on its underlying operation mechanism [16]; however, it is confirmed to contain a semiconductor-based radiation detection unit. The SGP sensor is controllable via the headphone jack of a smartphone (Figure 1a), when used in conjunction with the ‘Smart Geiger Pro’ app (available on the Apple and Android app stores). The app allows three measurement acquisition time options: 3, 5 and 10 min (also referred to as count times). After each measurement, the app displays values for the total counts, the number of counts per minute, and the dose rate (radiation delivered per unit time, μSv/hr) (Figure 1b). It is noted that it is not exactly known how the SGP-reported dose rate values are calculated by the sensor, however it is likely to be an estimate of equivalent or effective dose based on the counts measured. The dose rates were not independently verified, as absolute values of dose rate were not required for the purpose of the reported work. The app does not display an energy spectrum or allow for a specific energy peak selection. Therefore, lower energy gamma photons that have undergone Compton scattering cannot be distinguished by the SGP sensor. Measurements obtained with the SGP sensor were replicated by and compared with those acquired using a hospital-grade thyroid uptake sensor (Ortec Ace Mate); this is an inorganic scintillation crystal-based radiation sensor. Measurements of activity were made using a Capintec 15R dose calibrator; this is a device found in hospital radiopharmacies and is used to measure the activity of radioisotopes such as I-131.

### 2.2. Charactisation of SGP Sensor Performance: Influence of Treatment Parameters

#### 2.2.1. Decreasing Radioisotope Activity

During pretherapy measurements, the activity of the administered I-131 capsule held in the thyroid decreases exponentially (I-131 half-life = 8.02 days), and therefore the SGP sensor must be sufficiently sensitive to clinically relevant low activities. In order to ensure the SGP sensor had sufficient sensitivity for the decreasing I-131 activity, the dose rate was measured daily for a single I-131 source decaying over a period of 13 days. For these measurements, the SGP sensor was affixed to a retort stand located 20 cm from the I-131 source which was itself embedded in a Biodex Thyroid Uptake Neck Phantom (Biodex Medical Systems Inc., Shirley, NY, USA), as shown in Figure 1a. The cumulative dose was obtained using a count time of 5 min. Each measurement was repeated 3 times and an average dose rate (plus standard deviation) was then calculated. The dose calibrator was used to quantify the activity of the source.

To further characterize the SGP sensor sensitivity, the lowest activity which was detectable by the SGP sensor was estimated. Initially, a background measurement was recorded using the SGP sensor. It was then positioned at a constant distance of 10 cm from several low activity I-131 diagnostic capsules with activities between 0.0005 kBq and 0.057 kBq. The measurement count time was set at 5 min. The results are presented in Section 3.1.1.

#### 2.2.2. Measurement Reproducibility for Different Radioisotopes

To characterize the SGP sensor response to different radioisotopes yielding gamma photons of varying energy and thereby investigating its more widespread applicability, the SGP sensor response for different isotopes was investigated. This assessment was achieved by utilizing different isotopes, namely: Gd-153 (principal energy peak of 97 keV), Tc-99m (140 keV), I-131 (364 keV) and Cs-137 (662 keV). Each source had an activity between 4.8 MBq and 6.0 MBq. The 5 min acquisition time measurements were repeated 10 times for each isotope in order to calculate the mean number of counts detected, the standard deviation, and a coefficient of variability on the number of counts (standard deviation divided by the mean) and further investigate sensor response repeatability. The results are presented in Section 3.1.2.

#### 2.2.3. Scattering Material

Due to the natural variation in bodily composition between patients, a study to determine the SGP sensor’s response to different thickness of attenuating and scattering tissues was carried out. Sheets of 1 cm thick Plastic Water phantom were used as tissue mimicking material for the purposes of this investigation. A 4.50 MBq source of I-131 was inserted in the neck phantom and the distance between it and the SGP sensor was 15 cm. After each 5 min measurement, the amount of water phantom between the source and the sensor was increased by 1 cm up to a maximum thickness of 7 cm. The SGP sensor’s response to increasing layers of tissue mimicking material was compared to that of the standard thyroid uptake sensor (Figure 1c). The results are presented in Section 3.1.3.

### 2.3. Charactisation of Sensor Performance: Influence of External Parameters

#### 2.3.1. Sensor Motion

The influence of motion on the sensor accuracy was investigated, as motion may arise during patient monitoring at home, particularly if longer count/acquisition times are required. A series of measurements were taken while the sensor was first maintained in a fixed position affixed to a retort stand 15 cm from the neck phantom embedded with a capsule of I-131 (activity = 4.88 MBq). The same procedure was repeated while holding the phone by hand, again 15 cm from the I-131 capsule. The count time was set to 5 min for each measurement. This was repeated 6 times. The results are presented in Section 3.2.1.

#### 2.3.2. Sensor Positioning and Orientation

A similar procedure to Section 2.3.1 was utilized to determine the influence of sensor distance from the radioactive source and confirm that the SGP sensor obeys the inverse square law, i.e., for a point source, and in the absence of attenuation, the intensity of a beam of radiation will decrease as the inverse of the square of the distance from that source. The activity of the I-131 capsule in the neck phantom for these measurements was 2.86 MBq. The SGP sensor was fixed in a retort stand with an initial distance of 5 cm from the phantom. The distance was increased in increments of 5 cm after each 3 min measurement. Simultaneously, the same measurements were taken with the standard thyroid uptake sensor for comparison.

The influence of sensor orientation on sensitivity was also investigated. The surface showing the trefoil was denoted side A, and the surface displaying ‘Smart Geiger Pro’ was denoted side B. For these measurements, an I-131 source with an activity of 2.90 MBq was embedded in a neck phantom. The SGP sensor was affixed to a retort stand 10 cm from the phantom and was set to take 10 measurements, each of which were 3 min in duration. The results are presented in Section 3.2.2.

#### 2.3.3. Sensor Count Time

The three count time options (3, 5 and 10 min) were investigated to further ensure optimal patient home measurement parameters. The influence of count time was assessed with a 2.56 MBq source of I-131 again embedded in the neck phantom. The SGP sensor was again affixed to the retort stand positioned 15 cm from the source. A total of 8 measurements were made for each count time. The standard deviation and percentage variation were calculated for each count time and the results can be seen in Section 3.2.3.

### 2.4. Preliminary Patient Trial

Two sets of I-131 therapy patients were studied as part of this trial at St. James’s Hospital, Dublin: patients who had multiple dosimetry measurements prior to radioiodine therapy (Group A—multiple timepoint patients) and patients who had a single measurement prior to radioiodine therapy (Group B—single timepoint patients). All patients were being treated with radioiodine therapy for a benign thyroid disorder and would benefit from multiple dosimetry measurements, however Group B were unable to attend multiple dosimetry measurement appointments due to distance from the hospital. All patient measurements were monitored by at least one of the authors. Prior to the therapy, the RIU of the Group A patients was calculated each day after the initial diagnostic capsule was administered, for two/three days using Equation (1). These measurements were acquired simultaneously using both the SGP sensor and the standard uptake sensor; the SGP sensor was affixed to the standard uptake probe for the measurements for stability. Group B patients each had 3 min single measurements acquired again using both sensor types. Similarly, their RIU was calculated using Equation (1) and the results obtained from both methods were compared. The results for group A and group B patients are presented in Section 3.3.1 and Section 3.3.2.

## 3. Results

### 3.1. Charactisation of Sensor Performance: Influence of Treatment Parameters

#### 3.1.1. Influence of Decreasing Activity on Sensor Performance

Figure 2a shows the SGP-reported dose rate (µSv/hr) as a function of activity (MBq) of a decaying I-131 source as measured over a 13-day period using the SGP sensor; during this time the I-131 source activity is measured to linearly decrease from 8.01 to 2.01 MBq. The lowest activity detectable by the SGP sensor was found to be approximately 0.057 kBq for I-131. Measurements of I-131 capsules below this activity were indistinguishable from the initial SGP background measurement. This activity is much lower than that which would typically be administered for pretherapy measurements. A linear relationship between the SGP-reported dose rate and activity is observed (R^2^ value = 0.9907).

#### 3.1.2. Measurement Reproducibility for Different Radioisotopes

Figure 2b displays the coefficient of variation from 10 sequential 5 min measurements of the number of photons/counts for the different radioisotopes studied. For the lower energy isotopes (Gd-153 and Tc-99m), the measured mean count is low and the variation in the measured counts is relatively high (54% and 22%, respectively); I-131 (365 keV) showed the least variation in the number of counts (3%) among the range of isotopes studied.

#### 3.1.3. Influence of Scattering Material on Sensor Performance

Figure 2c shows the normalized number of counts measured by both the SGP and standard thyroid uptake sensors for different thicknesses of attenuating and scattering tissue mimicking material. As expected, the count was observed to decrease linearly with increasing thickness of attenuating material. The trend in normalized count as a function of attenuating material thickness is highly comparable for both sensor types.

### 3.2. Charactisation of Sensor Performance: Influence of External Parameteres

#### 3.2.1. Influence of Motion on Sensor Performance

The average count per minute was found to be 39.9 ± 2.9 when the SGP sensor was held in a fixed position, in comparison to 45.6 ± 5.7 when the phone and SGP sensor were hand-held. This corresponds to a variance in the repeated measurements of 7.3% and 12.5% for the fixed and hand-held SGP measurements, respectively. This implies that ideally the sensor should be placed or clamped in a fixed position before a measurement to improve accuracy. A simple frame device could be designed for use by patients at home for this purpose. However, it is worth noting that the two measurements agree within the stated uncertainties.

#### 3.2.2. Influence of Position and Orientation on Sensor Performance

The normalized counts obtained from both the SGP sensor and the standard thyroid uptake sensor were graphed as a function of the inverse squared distance from the source of I-131, and the results are shown in Figure 2d. An approximately linear relationship was observed for both sensors as expected, although some deviations were observed at the greatest distances tested (30 and 35 cm); a slightly higher R^2^ value was obtained for the standard thyroid uptake sensor (0.9928 vs. 0.9764), implying marginally better agreement with the inverse square law.

The influence of orientation of the sensor on the measured count rate was also investigated. Average counts per minute values of 69.7 ± 4.9 and 63.3 ± 6.3 were obtained when the sensor was orientated with side A and side B facing the source, respectively. The overlapping results indicate there is no statistically significant difference between the sensor sides when the calculated standard deviations are considered. The slightly reduced average count per minute value for Side B may be a result of increased attenuation due to the electronic configuration of the sensor.

#### 3.2.3. Influence of Count Time on Sensor Performance

The results for the influence of the count time on the sensor measurement are summarized in Table 1. The standard deviation and variation are shown to decrease as the count time is increased, as expected with any radiation detecting device.

### 3.3. Patient Trial

#### 3.3.1. Group A—Multiple Timepoint Patients

The RIU uptake percentages were measured and calculated for four pre-therapy dosimetry patients on days 1, 2 and 6 (patient 4 was unable to attend day 6) using both the SGP and standard thyroid uptake sensors. The results for patients 1, 2 and 4, shown in Figure 3, indicate that the two sensors performed comparably well, with relatively low uncertainties and discrepancies between the readings. However, there were significant variations observed for patient 3 in Figure 3c on days 1, 2 and 6. A possible cause of these large variations is excessive patient motion during the measurements; this is discussed further in Section 4.

#### 3.3.2. Group B—Single Timepoint Patients

Table 2 shows the RIU percentages as measured using the SGP sensor and the standard thyroid uptake sensor and the % difference between the two for each patient in Group B. A maximum % difference between the two sensors of 6% from the four patient measurements was observed.

## 4. Discussion

### 4.1. Charactisation of Sensor Performance: Influence of Treatment Parameteres

A pretherapy dosimetry study for I-131 uptake by the thyroid typically requires administration of an I-131 capsule with activity less than 10 MBq. This activity will then decay with time, and the measured dose rate will decrease correspondingly. In clinical use, a sensor is required to measure the I-131 time-activity curve within the thyroid, so that the I-131 treatment can be tailored to the patient’s specific I-131 biokinetics. It was essential therefore that the SGP sensor demonstrated its ability to accurately monitor a range of clinically relevant pretherapy activities.

The activity of the I-131 used for this study was 8.01 MBq initially. The dose rate as measured by the SGP sensor decreased linearly with decreasing activity of the decaying I-131 isotope (Figure 2a). The variability associated with the measurement of decaying I-131 was 7%, as determined from the fluctuations in the measurements. To further characterize the SGP sensor, the limit of activity detectability was also determined. For sources of I-131, the minimum detectable activity was estimated to be approximately 0.057 kBq. If a pretherapy patient is administered 0.5 MBq of I-131 (the lowest activity usually administered), 100% uptake on day 6 would equate to approximately 298 kBq, which is well above the limit of activity detection measured here for the SGP sensor. However, the % uptake of an I-131 capsule in a patient’s thyroid might be 50% (or lower, depending on their condition); this would equate to a remaining activity of 150 kBq on day 6. This is still well above the lowest activity detected by the SGP sensor, and verifies that the SGP sensor is suitable for operation in the activity range required for pretherapy dosimetry. However, a minimum starting I-131 activity may need to be specified for pretherapy dosimetry in order to minimize reading uncertainty, if it is validated for clinical use. A similar study was carried out by Brinks et al. [15] to characterize their collar detector, and a similar trend in accuracy of the number of counts measured was observed for an I-131 source. In their analysis, the counts measured by the collar detector was graphed as the activity of the source decreased. An average inaccuracy of 3% was calculated for the collar detector. It is important to note that the method used by Brinks et al. involved the measurement of a source that had an initial activity of 500 MBq, a typical activity a patient would be administered for therapy, as the aim of their study was to measure the retention curve of patients undergoing radioiodine therapy and not pretherapy dosimetry.

An assessment was then conducted with isotopes of varying principal peak gamma photon energies, in order to investigate the wider applicability of the SGP sensor in radionuclide therapy (Figure 2b). Before this assessment was done, it was expected that a linear relationship would occur between variance and peak photon energy, although this was not the case. The variance decreased with increasing gamma energies until a certain point (365 keV), after which it increased slightly. From this result, it is possible to infer that the SGP sensor has a window of photons energies at which optimum device performance is obtained and enough counts will be detected to minimize uncertainty. It is interesting to note that the SGP sensor shows relatively low measurement variability for the isotope Cesium (CS)-137. Cesium-137 has a variety of applications in radiotherapy and beyond; it is widely used for the calibration of radiation detection equipment (e.g., Geiger-Mueller counters).

Water phantom layers were then used to replicate the absorbing and scattering layers of tissues of the body. While tissue mimick thicknesses of up to 7 cm were studied, in reality the thyroid is very superficial and so approximately 2–3 cm would be the limit of clinically relevant thickness. The results for both the SGP and standard thyroid uptake sensor performance were compared (Figure 2c). It was found that the SGP sensor performed similarly to that of the hospital grade thyroid uptake sensor. This is an impressive result for the SGP sensor, considering that, unlike the standard thyroid uptake sensor, the SGP sensor has no energy windowing selection capabilities and so is unable to exclude scatter. Of course, I-131 has a relatively large principal gamma energy of 365 keV; the greater the gamma energy of the source, the more capable the source is of penetrating through increasing tissue. Nonetheless, it is important to verify that the SGP sensor shows the same response as the standard thyroid uptake sensor for patients of different bodily composition and size.

### 4.2. Charactisation of Sensor Performance: Influence of External Parameteres

As it is envisaged that the SGP sensor can be used by patients in their own homes in conjunction with a smartphone-based app, it was important to verify the influence of external factors such as motion and positioning on the sensor performance, and to establish the optimum set up parameters for patients to use the sensor unsupervised. The uncertainty associated with the sensor-phone being held by hand (12.5%) was greater than when it was fixed (7.3%), although not as much as may be expected. It should be noted that if the sensor was knocked while taking a measurement, it caused the count rate to spike. Therefore, it was concluded that a stand or support would ideally be supplied to patients to keep the sensor-phone stationary during measurements at home, but in the case that this is not possible, holding the phone with the sensor attached by hand will not cause excessive inaccuracy in measurements. The SGP sensor’s optimum directional orientation for measurements was also established. The results of this study indicated no statistically significant difference between the two sides of the SGP sensor. The slightly greater inaccuracy obtained for ‘side B’ may be due to the internal electronic configuration of the sensor.

It was important to verify that the SGP sensor obeys the inverse square law; this is a standard test conducted on all radiation detectors, to show that the device is behaving as expected. It may also be important in the context of patient remote usage of the SGP sensor in order to ensure that any separation in space between the patient’s thyroid and the SGP sensor (as may be required by the incorporation of a stand/support for the sensor) can be accounted for in calculation of patient RIU values. As expected, it was observed that the SGP sensor obeys the inverse square law (Figure 2d). The SGP sensor performed comparably to the standard thyroid uptake sensor, although the linear fit was determined to be slightly less accurate for the SGP sensor.

Finally, the influence of count time on the SGP sensor accuracy was investigated. It was concluded that the sensor performed more accurately as the count time was increased; only a 2% variation in the number of counts was observed for a 10 min measurement. Therefore, a count time of 10 min should be recommended to patients, when possible, to obtain the most precise data for RIU calculation. However, it is possible that multiple 10 min measurements a day may prove prohibitive for patients. In the case that a patient is not compliant, 5 min measurements can also be recommended to achieve a reliable result; a relatively low variation of 4% was determined for this shorter count time.

### 4.3. Patient Trial

A preliminary trial of use of the SGP sensor on four outpatients attending St. James’s Hospital for pretherapy dosimetry over multiple timepoints was conducted. The data was collected inside the hospital during outpatient appointments. The objective was to determine if the SGP sensor could replicate the data obtained by the standard hospital-grade thyroid uptake sensor, for real patients. The SGP sensor produced comparable results for patients who underwent pretherapy measurements for treatment of a benign thyroid disease (Figure 3, Table 2). However, a large uncertainty was observed in the percentage uptake values for patient 3 calculated using the SGP probe data; this was not replicated in the standard thyroid uptake probe data. It is probable that unlike patients 1, 2 and 4, patient 3 found it challenging to remain sufficiently still for the 3 min SGP sensor acquisition time. This is further supported by the fact that the discrepancy is repeated at the 2- and 6-day timepoint measurements (4 days apart), indicating that the issue is patient-specific. While the results in Section 3.2.1 show that any slight motion introduced by holding the SGP sensor by hand during measurements does not yield statistically significant differences in the sensor readings, it is possible that excessive patient motion above some threshold will prove problematic for use of the SGP sensor and may result in exclusion of certain patients from its use. Future studies will involve careful and systematic investigation of the influence of patient motion on the SGP sensor accuracy. Throughout testing, it was also found that the sensor recorded excess counts when knocked accidentally. This further supports the assertion that a stand or support is required to hold the sensor-phone stationary. Brinks et al. also conducted patient measurements with their collar detector for a source of I-131 with an activity of approximately 500 MBq [15]. To obtain the dose curves, they asked three patients to take six measurements at different times after the therapy. An inaccuracy of 8% was reported for the collar detector. By comparison, the SGP sensor as used for this study, gave an overall inaccuracy of 11% for only 3 measurements, when the outliers were not considered. It was intended that a greater number of patients be included, however the trial was cut short by the COVID-19 pandemic which has limited researcher access to the hospital. However, future work will include expansion of patient numbers as well as patient trialing of the device in their own homes.

While the results and analysis presented here illustrate the potential of the SGP sensor for use in pretherapy dosimetry, significant further work is required before such a device can be successful adopted for patient use. For instance, it must be established if the SGP sensor meets the definition of a medical device, i.e., it has a medical purpose and acts primarily by physical means. If so, then the risk class of the SGP sensor must be established, in accordance with the EU’s Medical Devices Regulations [18]. To facilitate this, a much larger validation study of the SGP sensor would be required. Within the EU, this is classified as a Clinical Investigation study and is used to assess the safety and/or performance of any medical device (both CE- and non-CE-marked). Once a device is correctly classified, it must then undergo its applicable conformity assessment procedure for it to be CE-marked.

## 5. Conclusions

The objective of this research was to fully characterize the function and performance of a commercially available smartphone-operated radiation sensor, with a view to assessing its suitability for clinical use in I-131 pretherapy dosimetry. The sensor was tested to study its accuracy, repeatability, and reliability to determine if it meets the requirements of an at-home radiation measurement device. It has been demonstrated that the sensor can accurately measure I-131 activities in the range that is relevant to pretherapy dosimetry. An activity of 0.057 kBq for I-131 was estimated to be the lowest detectable level. The suitability of the sensor for quantification of the dose rate of other high activity radioisotopes has also been shown.

It was shown that the commercial sensor performed comparably to a standard hospital-grade thyroid uptake sensor for both phantom test object devices with varying thicknesses of tissue mimick material, and indeed yielded comparable RIU measurements on real patients (*n* = 4). The influence of motion and orientation on the sensor performance was also investigated. It was concluded that while the device can be held by hand during use, ideally the sensor would be fixed in a support or stand at a fixed distance from the patient for the duration of the measurements to minimize errors introduced by motion; orientation of the sensor did not significantly influence the measurements. In order to maximize sensor accuracy, the longest count time permitted of 10 min should be used; however, the count time can be reduced to 5 min, if necessary, without a significant increase in error.

This work demonstrates the potential of low-cost commercially available radiation sensors as a solution for at-home pretherapy dosimetry for long distance patients, or indeed for hospitals who wish to implement dosimetry but do not have access to an uptake sensor.

Future work will include expansion of the preliminary clinical trial. Furthermore, due to the sensor’s strong performance in conjunction with high energy gamma photons, it’s suitability for application in positron emission tomography imaging and other radiotherapy applications will be explored.

## Figures and Tables

**Figure 1 sensors-22-09392-f001:**
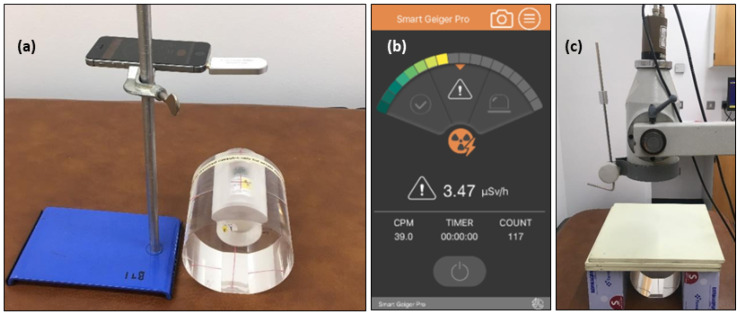
(**a**) Photograph of Smart Geiger Pro (SGP) radiation sensor, attached to smartphone via headphone jack, fixed by a retort stand at a distance of 20 cm from the I-131 source. The I-131 source is held in the anterior portion of a Biodex thyroid uptake neck phantom, simulating the position and resultant attenuation of activity from the thyroid gland in a patient’s neck; (**b**) Image of SGP smartphone app user interface; (**c**) Photograph of standard hospital-grade thyroid uptake Ortec sensor positioned vertically above the tissue-mimicking water phantom layers and the I-131 capsule embedded in the thyroid uptake neck phantom.

**Figure 2 sensors-22-09392-f002:**
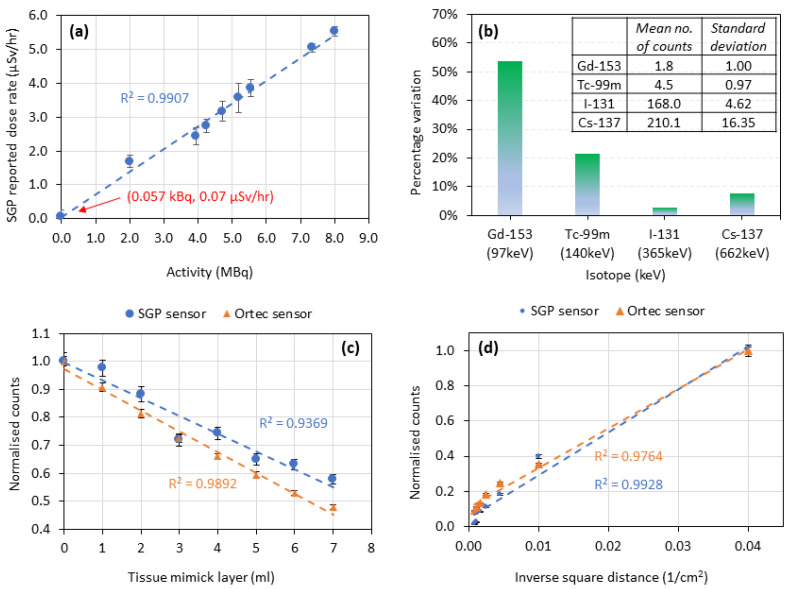
(**a**) SGP-reported dose rate (µSv/hr) vs. activity (MBq) for an I-131 capsule, measured using the SGP sensor. Dashed line represents a linear fit of the data (R^2^ value = 0.9907), and the data point indicated by the red arrow is the lowest I-131 activity detectable by the SGP sensor; (**b**) Coefficient of variation obtained from 10 measurements of the number of counts for different isotopes with various principal gamma energies (keV). Inserted table shows the mean number of counts and standard deviation for each isotope; (**c**) Normalised counts vs. tissue mimick layer/thickness (mL), measured using both the SGP and standard thyroid uptake (Ortec) sensors. Dashed lines represent linear fits of the data (R_SGP_^2^ = 0.9369. R_Ortec_^2^ = 0.9892); (**d**) Normalized counts vs. inverse squared distance (1/cm^2^) from the source of I-131, measured using both the SGP and standard thyroid uptake (Ortec) sensors. Dashed lines represent linear fits of the data (R_SGP_^2^ = 0.9764 and R_Ortec_^2^ = 0.9928). The error bars on all data points represent the standard deviation of three independent measurements.

**Figure 3 sensors-22-09392-f003:**
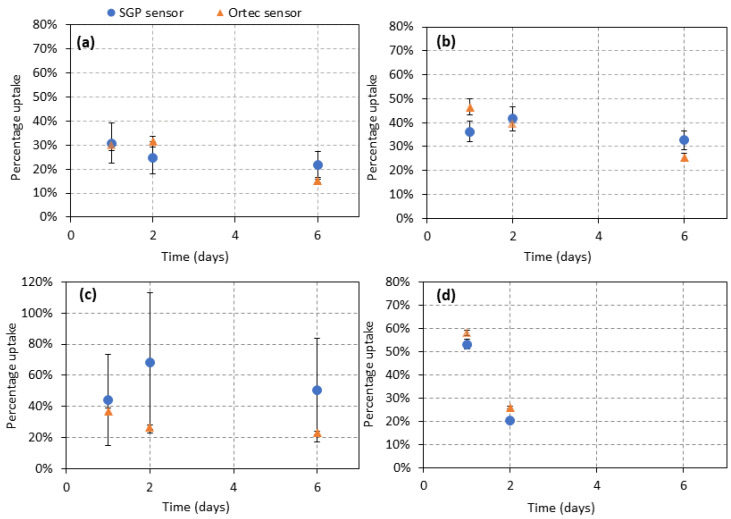
Group A RIU values of as a function of time (days), for (**a**) Patient 1, (**b**) Patient 2, (**c**) Patient 3 and (**d**) Patient 4. The error bars on all data points represent the standard deviation of two independent measurements.

**Table 1 sensors-22-09392-t001:** Average counts per minute (CPM) measured for a 2.56 MBq I-131 capsule and associated error for each SGP sensor count time option.

Time	Average CPM	Standard Deviation	Variation
3 min	37.4	2.9	8%
5 min	35.8	1.3	4%
10 min	36.3	0.6	2%

**Table 2 sensors-22-09392-t002:** Group B patients uptake percentages given by both sensors, and the difference between the two.

Patient	Ortec Sensor	SGP Sensor	Difference
Patient 5	41%	47%	6%
Patient 6	48%	51%	3%
Patient 7	30%	34%	5%
Patient 8	41%	54%	4%

## Data Availability

Not applicable.

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
