# Peer review of "Investigation of the Suitability of a Commercial Radiation Sensor for Pretherapy Dosimetry of Radioiodine Treatment Patients"

_sensors, 2022, doi:10.3390/s22239392_

Round 1

Reviewer 1 Report

Specific comments:

* Line 75 "The radioactivity of the diagnostic capsule is then measured intermittently..." - I suppose the activity of the THYROID gland is meant here...? please explain

* Linie 82 "where the net counts administered reference value are obtained by measuring the ‘counts’ or gamma photons emitted from a standard I-131 capsule... " - one should think that the reference counts (in the denominator of eqn 1) are the counts obtained before administration from the capsule being administered for treatment, not some arbitrary standard capsule; please explain.

* Equation 1 units: it seems 'counts' is the correct unit for the Ortec probe, but the SGP seems to produce a reading in uSv/hr, not counts. The authors should explain this in the text (otherwise leads to confusion e.g. in Figure 1.a y-axis units). 

* Line 84 "The obtained dose profile as a function of time..." - shouldn't that more fittingly be called a COUNT PROFILE, rather than a dose profile? - please explain. 

* Should include a more precise model number of the sensor, e.g. SGP-001. 

* Should include a picture of the SGP sensor and it's positioning 

* Should include picture of pertinent SGP software screen shot

* Line 131 "the app displays the dose rate..." - should this be DOSE rate or COUNT rate; please explain. Note: if the app does indeed display dose (i.e. radiation dose) rate, comment that this is unrealistic/unreliable; if the app displays counts, please change the text to say COUNT RATE. 

* Line 142 "...During pretherapy measurements, the activity of the administered I-131 capsule will decay exponentially..." - this is not clear; once administered, there IS NO MORE capsule; it will be dissolved in the patient and the contents distributed through the body; please clarify and explain. 

* Line 143 "...the SGP sensor much be..." change to "...the SGP sensor MUST be "

* Line 151 "minimum detectable activity" - MDA is usually a precisely defined term; please give precise reference how MDA was determined/calculated or present calculation/equation in the paper; if MDA was not exactly established, but rather just handwavingly estimated with the test sources, please state that explicitly.  

* Linearity (Fig. 1.a) was determined starting from ~8 MBq, down to only ~4 MBq; this seems to be a clinically insufficient range, since one would expect activity in the patient thyroid to reach values much below 4 MBq over time. This should be extended down to the lowest clinically expected values (background?). Also, measurement error for each individual measurement should be reported (error bars) or at least be commented on if not available. Also, the y-axis label should make it clear that this is only the dose rate REPORTED by the device, and therefore questionable; please change to something like 'SGP-reported dose rate (uSv/hr)'.

* Line 198 "...influence of sensor orientation on performance..." - the use of the word 'performance' here does seem ambiguous; do the authors mean the influence on SENSITIVITY? Please clarify

* Section 2.1.2 Photon energy - the heading and the first half of this section lead the reader to believe that relative sensitivity between energies (isotopes) is being discussed. Only in the second half is it revealed that actually reproducibility is tested for different energies (isotopes). Please change the heading to something like 'Reproducibly for different isotopes'. Also, please define 'coefficient of variability' at least verbally, i.e. something like 'standard deviation divided by the mean for each isotope' or whatever the case may be; even better would be an (in-line) equation.

* Related to previous, in Fig 1.b: not good to show percentages; better to show actual measured means with error bars for variability (e.g. std.dev.); can include numerical values for reproducibility as defined as per previous comment right above.

* Figure 1.c should include error bars for each measurement

* Figure 1.d should be semi-logarithmic and also contain error bars. 

* Table 1. heading - should mention the activity of the I-131 capsule being used for this test (so the reader won't have to search for it in the text) 

* Line 255 "... variance in the repeated measurements, along with the associated error, increase when the phone and sensor are held as opposed to set in a fixed position..." - please calculate the variance for each test and quote in the text e.g. as percentage. Also, the two results agree within uncertainty, which should also be mentioned in the text.

* Line 256 "...sensor are held..." - did the authors mean HAND-HELD (as opposed to fixed position)? Please clarify the difference between 'held' and 'in  a fixed position'

* The heading for section 3, RESULTS, is missing

* Figure 2 should contain error bars for each data point.

* Figure 2.c, the outlier: the authors explain this with excessive motion; but the 2hr and 6 hr SGP time points show the same trend as the Ortec data. Do the authors suspect similar excessive motion at both time points? Also the effect of motion has been tested (sec. 3.2.1.) and found not to be significant. Are the authors able to re-produce deviations as large as shown in Fig. 2.c. with their tests? Perhaps do more tests with greater motion to try and reproduce this? I believe more care needs to be taken to explain this outlier; motion alone does not seem to be the cause. Could not a wrong measurement distance be the cause for this systematic problem? Has this been investigated? The authors should comment more on this because if these remain unexplained, confidence in their method will inevitably suffer. 

* Line 330 " If a pretherapy patient is administered 0.5 MBq of I-131 (the lowest activity usually administered), uptake on day 6 would equate to approximately 298kBq,..." - this is only a calculation; but tests of the SGP device have not been carried out down to these levels; this should be done, e.g. for linearity and others,  in order to  verify its for the intended purpose. 

* Line 377 "...holding the phone with the sensor attched may not cause excessive inaccuracy in measurements..." - this seems to contradict what the authors concluded for the outlier in Figure 2.c. Please clarify and refine discussion of the outlier (see previous comment above)

General comments: 

* An interesting idea worthy of a manuscript; it's nice to see commonly available technology being made use of in this way. However, discussion should include the regulatory obstacles to serious clinical adaptation.

* The authors generally seem to struggle with units for quantities, e.g. not consistently distinguishing between counts and dose; care should be taken to achieve consistency throughout the manuscript. 

* Absolute calibration of the device has not been checked by the authors; i.e. how accurate does the SGP report doses (uSv/hr)? I would think dosimetric follow-up is possible by relative measurements (% uptake), using some measurement (e.g. with a bioassay probe) for verification and calibration as needed, as the authors have done. The manuscript should clearly point out that even though the SGP reports dose rate (uSv/hr), this has not been verified and also is not needed for the purposes of this paper. 

* The manuscript lacks some rigor, e.g. in determining minimum detectable activity (MDA), or in determining linearity (Fig. 1.a) which should be verified down to much lower activities. 

* I certainly recommend for publication after gaps in methodology and presentation have been filled. 

Reviewer 2 Report

Reviewer report

MS title: Investigation of the ...

MS ID:   MDPI Sensors

Authors present the evaluation of a radiation detector operated with a smarthphone aiming to measure the dose in the pre-treatment with 131I.

The work is original and easy of follow. 

The topic is in the aims of MDPI Sensors 

journal and deserve to be published.

Author Response

We thank the reviewer for their time and their positive feedback.

Reviewer 3 Report

O’Callaghan et al. characterized the function and performance of a commercially available smartphone-operated radiation sensor, with a view to assessing its suitability for clinical use in I-131 pretherapy dosimetry. The sensor was tested to study its accuracy, repeatability, and reliability to determine if it meets the require ments of an at-home radiation measurement device. The sensor can accurately measure I-131 activities in the range that is relevant to pretherapy dosimetry. 

So far, I donot think it can be published in Sensors because I did not see  new concept or new way of thinking. If you can provide more evidence of the novelty of the manuscript, rather than primarily reporting technological improvements, I will reconsider it.

Author Response

We thank the reviewer for the opportunity to respond and provide evidence for the novelty and potential impact of the reported work. In fact, in this work we are not reporting, or claiming to report, technological improvements; this manuscript presents the novel and non-obvious use and characterisation of a commercially available device for a highly impactful application: pretherapy dosimetry of I-131 treatment patients.

The two key areas of novelty in this work are:

  1. Characterization of a radiation sensor for pretherapy radioiodine dosimetry applications

Previous research efforts in the field of radiation sensors in I-131 treatment have been described in the Introduction section of the manuscript. These prior works have focused on wearable sensors for I-131 therapy monitoring during therapy, not for pretherapy dosimetry measurements. During therapy, patients are administered significantly higher activity capsules of I-131 (500 MBq) in comparison of pretherapy measurements (0.5-5MBq), and therefore while these works are important for background, the sensor requirements in terms of performance, sensitivity etc. are entirely different for pretherapy. This work represents, to the best of our knowledge, the first detailed systematic study of a radiation sensor specifically for pretherapy applications. The following changes have been made to the text of the introduction to highlight the difference to prior work and emphasise the novelty of this manuscript:

Lines 96-98: “In recent years, efforts have been made to develop wearable radiation detectors to enable remote monitoring of I-131 uptake in the thyroid following administration of the I-131 therapy; however, no studies to date have focused on pretherapy dosimetry itself.”

Lines 101-103: “In (Brinks’ et al [15]) study, the novel measurement device was used during therapy (initial administered I-131 activity of 500 MBq) to measure the patient’s I-131 uptake curve and the counts per second were displayed on an associated handheld tablet.”

Lines 108-110: “As in [15], the intended purpose of this device is to monitor the real-time uptake of I-131 by the thyroid, once the I-131 treatment has been administered, rather than for pretherapy dosimetry applications.”

Lines 123-124: “To the best of the authors’ knowledge, this is the first time a feasibility study of a radiation sensor for pre-I-131 therapy dosimetry has been reported.”

We have also endeavoured to emphasise the regulatory need for pretherapy dosimetry for the general reader. This has been more thoroughly described in lines 55-57 of the revised manuscript:

 “It [pretherapy dosimetry] is also mandated by the Basic Safety Standards Directive of the European Union [9]. Despite this, there is no widespread uptake of this approach among centers offering I-131 therapy for benign thyroid diseases.”

  1. Investigation of a non-collar based radiation sensor for pretherapy dosimetry

A key factor in the lack of adoption of pretherapy dosimetry has been the lack of suitable technologies for patients to measure their thyroid activity levels remotely. The two-prior works on I-131 radiation sensors (for therapy, not pretherapy as is the case here) described in the introduction section are both collar-based devices which many patients find uncomfortable. As per line 104, “While the proposed (collar-based) measurement device was deemed feasible for such measurements, patients reported discomfort while wearing the collar for extended periods”. Patient comfort is essential if patients are to reliably use the radiation measurement device in their own homes, away from the supervision of medical professionals. The SGP sensor proposed here for pretherapy dosimetry overcomes this issue as it is handheld and lightweight. We have shown (section 3.2.1) that holding the sensor by hand during the measurement does not significantly impact the reliability of the measured results, and thus a collar is not required.

In order to clarify the novelty of this non-collar based approach further for the reader, the following text has been edited/added:

Lines 104-106: “While the proposed collar-based device was deemed suitable for such measurements, patients reported discomfort while wearing the collar for extended periods. This suggests that a patient self-monitoring device should ideally be small, handheld, and lightweight.”

Lines 116-118: “Furthermore, the handheld nature of the (SGP) device, rather than a wearable collar, ensures minimal patient discomfort during the measurement, which is crucial to maximize patient compliance.”

 This manuscript provides a novel potential solution for pretherapy dosimetry: a low cost, handheld and readily available device which can be conveniently used by patients in their own homes. The feasibility of use of the device for pretherapy dosimetry has been extensively characterised. This work is a non-obvious, novel and 'inventive step'; this ensures it meets the requirements for novelty required in academic publishing and indeed patentability under most patent laws worldwide. The low-cost and commercial availability of the SGP sensor is crucial to enable its rapid adaptation by medical professionals in a variety of settings. This work has been deemed suitably novel and interesting to the readers of MDPI Sensors by three other independent peer-reviewers. We hope that we have convinced the reviewer of the merit of this work. 

Reviewer 4 Report

This ms is very well segmented and written. It was pleasure to read it and I recommend it for publication.

I have few remarks.

Line 131 Written “dose rate”. It is not clear which dose do you mean.

Later in the text, term “dose” was used frequently. Again, the same question “which dose” (absorbed, equivalent or smth else).

Some scheme of measuring setup or even an image (phantom, source - detector  position) will be very welcome.

Round 2

Reviewer 1 Report

The manuscript has been much improved and I commend the authors for responding to all queries and incorporating the suggested changes.

A last, minor, comment is that the activity of the lowest data point in Figure 2(a) (linearity) is visually indistinguishable from zero. The authors should explicitly mention its value (I believe 0.057 kBq?) either in the figure caption or in the chart area.

Other than that, I am satisfied that the manuscript is now ready for publication.

Author Response

We thank the reviewer for their suggestion. To address this, the (x,y) coordinates of the data point corresponding to the lowest activity detectable by the SGP sensor (0.057 kBq, 0.07 uSv/hr) have been added to figure 2(a) in the revised manuscript. The caption of figure 2(a) has also been amended to include the following (line 298): "the data point indicated by the red arrow is the lowest I-131 activity detectable by the SGP sensor".

Reviewer 3 Report

I think it can be accepted.

Author Response

We thank the reviewer for their opinion that the manuscript can now be accepted.